# Control of Organ Abscission and Other Cell Separation Processes by Evolutionary Conserved Peptide Signaling

**DOI:** 10.3390/plants8070225

**Published:** 2019-07-15

**Authors:** Chun-Lin Shi, Renate Marie Alling, Marta Hammerstad, Reidunn B. Aalen

**Affiliations:** Section for Genetics and Evolutionary Biology, Department of Biosciences, University of Oslo, N-0316 Oslo, Norway

**Keywords:** IDA/IDL, abscission, cell separation, peptide signaling, abscission zone

## Abstract

Plants both generate and shed organs throughout their lifetime. Cell separation is in function during opening of anthers to release pollen; floral organs are detached after pollination when they have served their purpose; unfertilized flowers are shed; fruits and seeds are abscised from the mother plant to secure the propagation of new generations. Organ abscission takes place in specialized abscission zone (AZ) cells where the middle lamella between adjacent cell files is broken down. The plant hormone ethylene has a well-documented promoting effect on abscission, but mutation in ethylene receptor genes in *Arabidopsis thaliana* only delays the abscission process. Microarray and RNA sequencing have identified a large number of genes differentially expressed in the AZs, especially genes encoding enzymes involved in cell wall remodelling and disassembly. Mutations in such genes rarely give a phenotype, most likely due to functional redundancy. In contrast, mutation in the *INFLORESCENCE DEFICIENT IN ABSCISSION* (*IDA*) blocks floral organ abscission in Arabidopsis. *IDA* encodes a small peptide that signals through the leucine-rich repeat receptor-like kinases HAESA (HAE) and HAE-LIKE2 (HSL2) to control floral organ abscission and facilitate lateral root emergence. Untimely abscission is a severe problem in many crops, and in a more applied perspective, it is of interest to investigate whether IDA-HAE/HSL2 is involved in other cell separation processes and other species. Genes encoding IDA and HSL2 orthologues have been identified in all orders of flowering plants. Angiosperms have had enormous success, with species adapted to all kinds of environments, adaptations which include variation with respect to which organs they shed. Here we review, from an evolutionary perspective, the properties of the IDA-HAE/HSL2 signaling module and the evidence for its hypothesized involvement in various cell separation processes in angiosperms.

## 1. Introduction

Cell separation processes, especially abscission, are in general of great importance in plants, in relation to reproduction, development and adaptation to the environment, but the sites of cell separation differ between species. *Arabidopsis thaliana* sheds individual floral organs (petals, sepals, and stamen) shortly after pollination [1], and displays opening of anthers and siliques (so-called dehiscence) to release pollen and drop seeds [2]. Dehiscence involves specialized tissues that breaks open when tension builds up upon dehydration [3]. The loss of organs is achieved by degradation of the middle lamella between adjacent cell files in specialized abscission zones (AZs) at the base of the organ to be shed. The abscission process can be divided in four stages, namely formation of the AZ, conditioning of the AZ so it can respond to signaling molecules, cell wall loosening followed by the actual cell separation and finally generation of a protective layer [2]. In contrast to many other species, Arabidopsis does usually not shed cauline leaves, whole flowers, fruits or leaves; however, environmental stress factors like drought can induce abscission of cauline leaves [4]. Cell separation events are also found below ground, facilitating lateral root emergence and releasing root cap cells [5,6].

During the last couple of decades the invention of technologies for genome-wide determination of gene expression levels, such as microarrays and, later, RNA sequencing, have facilitated a transcriptomic approach to understand these processes. This approach gives unbiased access to the gene expression pattern in a given organ at a given time. However, organs consist of several tissues, and such experiments do not disclose more specific expression patterns within an organ. Furthermore, genes with low expression levels, which is typical for signaling molecules, can easily be overshadowed by more abundantly expressed genes, which are not necessarily the most important factors with respect to control of whole processes. The genetic approach, that is, the identification of mutants impaired in the relevant process, can pick up important genes with low activity. From this perspective we review the evidence for the importance of the signaling peptide IDA and its receptors in different cell separation processes both above and below ground in Arabidopsis, and also in abscission of diverse organs in a variety flowering plants.

## 2. The Role of IDA/IDL Peptides during Organ Abscission and Other Cell Separation Processes

### 2.1. Control of Cell Separation Processes by IDA and IDA-LIKE Peptides in Arabidospsis

#### 2.1.1. Mutants Deficient in Abscission

Few mutant blocking cell separation processes have been identified. The *bop1 bop2* (*BLADE-ON-PETIOLE1/2*) double mutant in Arabidopsis and the *jointless* mutant in tomato fail to develop AZs at the base of the floral organs and at the base of the pedicel, respectively [7,8,9]. This indicates that organ abscission is dependent on specialized cells. Mutations in the Arabidopsis genes *NEVERSHED* (*NEV*) and *INFLORESCENCE DEFICIENT IN ABSCISSION* (*IDA*) result in permanently attached floral organs [10,11,12]. The *nev* mutant is impaired in the golgi apparatus, and most likely fail to export a number of proteins needed for middle lamella dissolution in the AZ [11].

*IDA*, on the other hand, is member of a small family of 9 genes encoding IDA and IDA-LIKE (IDL) preproproteins of less than 100 amino acids with an N-terminal export signal, a variable part, and near the C-terminus, a commonly conserved motif of 20 amino acids named the Extended PIP motif (EPIP) [10]. By analogy to CLAVATA3 (CLV3), one of the first signaling peptides identified, shown to be a ligand of a leucine-rich repeat (LRR) receptor-like kinase (RLK) (CLAVATA1), we anticipated that IDA could be a ligand of LRR-RLKs. Of the approximately 235 LRR-RLK genes in Arabidopsis, *HAESA* (*HAE*) and the closely related *HAE-LIKE2* (*HSL2*), were found to be expressed in the floral organ AZs, and the *hae hsl2* double mutant was, like *ida*, totally deficient in abscission (Appendix A) [13,14].

The abscission process can be monitored by measuring the force needed to pull of the petals, the so-called petal breakstrength (pBS) [15]. In wild type Arabidopsis, the high pBS at anthesis is gradually reduced to zero due to cell wall loosening, and at position 8 (counting flowers from the top of the inflorescence) the organs are lost [13]. The *ida* and *hae hsl2* mutants initially display a similar, but delayed pBS profile, thus indicating that IDA is regulating some cell wall remodelling (CWR) genes. For example, expression levels of XYLOGLUCAN ENDOTRANSGLYCOSYLASE6 (XTR6), POLYGALACTURONASE LATERAL ROOT (PGLR) and POLYGALACTURONASE ABSCISSION ZONE ARABIDOPSIS THALIANA (PGAZAT) are greatly reduced in *ida* and *hae hsl2* mutants during floral organ abscission [5], consistent with microarray and transcriptome data [16]. However after position 10 the pBS increases again to the level at anthesis [13].

In a recent reevaluation of the literature on the role of IDA in abscission it was suggested that the lack of detachment of the floral organs in these mutants was due to lowered turgor pressure resulting in failure of expansion of AZ cells [17]. However, the floral organs are in fact turgid when the pBS value is at its lowest (position 10), and normally remain like that at least up to position 16 (Appendix A). Furthermore, when exposed to ethylene the floral organs of both the *ida* mutant and wild type senesce, but only in the mutant the floral organs remain attached. Conversely, the *nev* mutant show deficiency in abscission although the AZ cells expand [10,12]. Thus, there is no strict correlation between turgor loss and failure to abscise, and also no strict correlation between cell expansion and abscission.

#### 2.1.2. IDA and Induction of Abscission

The ability of IDA to induce abscission has been questioned [17]. However, overexpression of IDA, using the constitutive 35S promoter, leads to earlier abscission of petals, sepals and filaments, as well as ectopic abscission at the base of the pedicels, branches and cauline leaves [13]. These are sites where abscission takes place in many other species, and in Arabidopsis addition of IDA is sufficient for induction of abscission. Thus, the other conditions and factors needed for abscission to take place must be present. Consistent with this, vestigial AZs have been discovered at these sites, and the HAE and HSL2 receptors are expressed there [13,18]. Three peptides with similarity to IDA, IDA-LIKE6 (IDL6), IDA-LIKE7 (IDL7) and IDA-LIKE8 (IDL8) (Appendix A), are involved in biotic and abiotic stress defence [19], however, accumulating evidence suggests that signaling peptides can be involved both in development and defense [20]. Drought stress may, for example, induce IDA expression at the base of Arabidopsis cauline leaves, leading to abscission of these organs [4]. The fact that all cells are not falling apart from their neighboring cells in the *35S:IDA* lines points to the requirement of conditioned cells that are responsive to IDA signaling. The *35S:IDA* phenotype is dependent on functional HAE and HSL2 [13]. Biochemical and structural evidence has substantiated IDA as a ligand for HAE and HSL2 [21]. Activation of HSL2 results in an oxidative burst, and an assay where synthetic IDA peptides of different concentrations and lengths were added to tobacco leaf pieces expressing HSL2 demonstrated that the 12 amino acids of PIP with hydroxylated proline (Pro, P) in position 7 could bind and activate the receptor efficiently at biologically relevant nanomolar concentrations, hundred times more efficiently than EPIP [21]. The histidine (His, H) and asparagine (Asn, N) in positions 11 and 12 seem essential for function, but the amino acids C-terminally to Asn can be removed.

Synthetic hydroxylated IDA PIP peptide (Figure 1) bound to the HAE-LRR was used to obtain the molecular structure of this ligand-receptor pair and their interaction with the co-receptor SOMATIC EMBRYOGENESIS RECEPTOR KINASE1 (SERK1) [22]. In vitro studies have shown that subtilases can digest IDA two amino acids upstream of PIP, suggesting that the peptide is 14, not 12 amino acids long [23]. So far, the active peptide present in planta has not been isolated. Although the expression patterns of *IDL* genes differ from that of *IDA*, overexpression of *IDL1-5* resulted in phenotypes very similar to that of *35S:IDA* [13]. This suggested that IDL peptides could signal through the receptors of IDA. However, quantity may overshadow low quality binding, as it turned out that only *IDL1* could complement the *ida* mutation when under control of *IDA’s* own promoter [13]. This indicates that the residues of particular functional importance are those that differ between IDA or IDL1 and IDL2-5, that is Pro position 4 and Arg position 10 which have been changed to alanine or serine (Ala, A; Ser, S), and lysine (Lys, K), respective (Figure 1). At present, receptors of IDL2-5 have not been identified.

Further dissection of the *IDA* gene showed that the variable part and the C-terminal end are not necessary for function, however, plants transformed with *35S:IDA* constructs without the N-terminal export sequence do not display the overexpression phenotypes [13]. The prepropeptide is thus assumed to be exported to the endoplasmic reticulum, processed, and secreted to the apoplastic space where it can move and interact with its receptors present in the plasma membrane.

#### 2.1.3. IDA and IDL1 Are Involved in Cell Separation Events in the Root

We have discovered that *IDA* is also expressed below ground, in the cells overlying emerging lateral roots (LR) [5]. LRs are initiated in the pericycle in Arabidopsis; the LR primordia (LRP) have to pass three cell layers, endodermis, cortex and epidermis, to emerge (Figure 2). Mutations in *IDA*, *HAE* or *HSL2* delay LR emergence, since the overlaying cells do not separate to let the lateral root primordia (LRP) out. However, these mutants do not stop the primordia from growing, and in *ida* and *hae* mechanical force from the growing LRP seems responsible for localized destruction of cells directly overlaying the emerging LRP [5]. Transcription factors working downstream of IDA-HAE/HSL2 during floral organ abscission and cell wall remodeling genes, typically expansins (EXPs), xyloglucan endotransglucosylase/hydrolases (XTHs), and PGs have been identified through suppressor screens and mutant studies [24,25]. Interestingly, a number of the same factors have been found to be involved in separation of cortex and epidermal cells to facilitate LR emergence [5,26].

Another cell separation event found in the Arabidopsis root is sloughing of root cap cell layers that cover the root tip with the meristematic zone, the stem cell niche surrounding the Quiescent Centre (QC) and the gravity-sensing columella cells [27] (Figure 2). *IDL1* is expressed in the cap, which is protecting stem cells from damage and sensing environmental cues [13]. The outermost layer of the cap is detached at regular intervals. We have found that *IDL1* and *HSL2* are expressed at the tip, and, using the same methods as for IDA, it could be shown that IDL1 efficiently binds and activates HSL2 [6].

One might expect, in analogy with the floral mutant phenotype, that the *idl1* and *hsl2* mutants would accumulate root caps due to lack of cell separation. Instead the number of attached root cap layers was maintained, but the whole process, with maturation of columella and lateral root cap cells leading to the detachment of the outermost layer, and generation of new cell layers, was slowed down [6]. Enhanced expression levels of *IDL1* generated more sloughed root caps, but the number of attached layers was still maintained. In contrast to other cell separation processes, root cap sloughing is a recurrent event in the same organ, and therefore it is necessary to compensate every sloughed layer by the generation of a new. In the mutants, the frequency of sloughing was significantly reduced, as fewer caps were detached.

From an evolutionary perspective, we find it interesting that seemingly very different developmental processes, above and below ground, employ the same molecular mechanism (Figure 2).

### 2.2. Involvement of IDA/IDL1 in Abscission Processes in Other Flowering Plants

#### 2.2.1. *IDA/IDL1* and *HAE/HSL* Genes Are Present in All Angiosperm Orders

We have undertaken a phylogenetic investigation to identify how widespread the IDA/IDL-HAE/HSL signaling module is in species other than Arabidopsis [18]. To investigate the evolutionary history of the IDA/IDL-HAE/HSL2 signaling module, we collected putative HAE/HSL2 and IDA/IDL sequences from The National Center of Biotechnology Information (NCBI), Phytozome and Comparative Genomics (GoCe), using the Basic Local Alignment Search Tool (BLASTp and tBLASTn). HSL2 orthologues were found in basal angiosperms, mono- and dicot orders, with the exception of the grasses (Poales), while HAE is the likely result of a whole-genome duplication in the dicot linage before the split into Rosids and Astrids [18].

We checked that potential IDA hits represented prepropeptides of about 100 amino acids with an N-terminal hydrophobic export signal and the PIP motif near the C-terminus. Analysis of the PIP motif in more detail identified three prototypes that were named after the position 10 residue, PIP_R_ representing IDA/IDL1, PIP_K_ representing IDL2-5, and in addition, PIP_Q_ in particular found in the grasses (Poales) (Figure 1) [18]. We concluded that the PIP_R_ motif has been conserved over the 175 million years flowering plants have evolved, as it was present in all orders from the most primitive to the most advanced angiosperms. Only two out of the twelve amino acids in the PIP_R_ motif differ over evolutionary time, valine (Val, V) or isoleucine (Ile, I), which are found in position 2 and Ala or glycine (Gly, G) in position 6. In both cases the alternative amino acids have similar properties. The PIP_K_ variant could be followed back at least to the early eudicots [18]. IDL4, which, in addition to Lys10, is characterized by mutation in the Pro in position 4 (Figure 1), most likely arose after an early whole genome triplication before the appearance of the Vitis order [28].

Further analysis of the sequences collected revealed a correlation between the amino acid in position 10 and the amino acid motifs present on each side of the PIP motif, for example, the common amino acid C-terminally to PIP is Ser Val (SV) in IDA/IDL1 propeptids, but DIG in IDL2-5 (Figure 1). We assume that these are recognition sites for processing enzymes. There are 54 genes encoding subtilases in Arabidopsis and three of these, *SBT4.12*, *SBT4.13* and *SBT5.2* have been shown to process IDA between Lys and Gly in the conserved motif LPKG two amino acids upstream of PIP_R_ (Figure 1) [23]. The IDL propeptides have different signatures, suggesting that they are processed by other subtilases. Typically, each species has one or two genes encoding PIP_R_ IDA peptides, and beside this, IDL4 PIP_K_ variants. Some of these genes are the result of more recent genome duplications, like IDL1, IDL2 and IDL3, which are specific to Brassicae.

#### 2.2.2. Expression Data and Functional Testing Suggest IDA/IDL Regulation of Abscission of Diverse Organs Both in Dicots and Monocots

To substantiate not only the presence of *IDA/IDL* and *HAE/HSL* genes in the genomes of flowering plants, but also their potential involvement in cell separation events, identification of relevant expression patterns is needed, such as expression in AZs of abscising organs (Table 1). Over the last few years when RNA sequencing and transcriptome analyses have become fast and economically affordable methods, gene expression patterns in AZs versus non-abscission zones have been investigated in many species, for example fruit trees like citrus species and litchi where abscission is a problem (Table 1). Such analyses reveal that similar CWR genes are transcribed, even when in different organs [29,30,31,32,33]. Compared to that of enzyme-encoding genes, transcription of signaling peptide genes may be less abundant and may therefore be overlooked, and additionally the focus has often been directed more towards traditional hormones.

One example is the recent transcriptome investigation of the AZs of yellow lupine (*Lupinus luteus*), belonging to legumes under the order Fabales of the Rosids. Lupine seeds are used as animal feed and a source for vegetative oil in parts of the world. Due to their high protein content, lupine seeds represent a significant alternative to soybean (*Glycine max*), but premature abscission of flowers is a problem that affects yield.

The whole genome sequence of yellow lupine is not available, but RNA seq has been performed on AZ and non-AZ tissue, as well as AZ in flowers of different developmental stages, and transcriptional differences have been investigated with an explicit focus on cell wall modifications and hormone metabolism [31]. However, data on *IDA*, *IDL* and *HSL* transcript levels could be extracted from the project (BioProject ID PRJNA419564) by tBLASTn searches against the sequenced RNA using Arabidopsis HAE/HSL2 or the prepropeptides of IDA/IDL or as queries. Transcripts representing two putative lupin PIP_R_ orthologues (named *LlIDA* and *LiIDL1*) and three PIP_K_ orthologues (*LlIDL2*, *LlIDL3* and *LlIDL4*) as well as putative orthologues of *HAE* and *HSL2* were identified (Figure 3a). *LlIDA* and *LlHSL2* showed higher expression levels at the base of pedicels with rather than without developed AZ (Figure 3b), and the expression levels in AZs in developing flowers increased from green buds to the post-pollination stage (Figure 3b,c). These data indicate that the lupine gene most similar to *AtIDA* plays the most important role in lupine AZs compared to the *LlIDL* genes, however, this has not been confirmed by other methods, such as qPCR.

The cDNA for one of the PIP_K_ type genes with moderate expression levels compared to *LlIDA*, was recently cloned [40]. Application of 10–100 µM synthetic EPIP motif for this *LlIDL* gene on non-abscising AZ increased the number of abscised flowers. These concentrations are, however, comparable to overexpression using the 35S promoter, and significantly higher than normal biological levels. When considering the published Arabidopsis data on the efficiency of different AtIDA/AtIDL peptides [21], it is not clear why an EPIP motif from an *LlIDL* gene and not a PIP_R_ motif from an *LlIDA* gene was chosen for this experiment.

Expression of *IDL* and *HSL* homologues have also been investigated in soybean, belonging to the order Fabales of the Rosids, which has four PIP_R_ and eight PIP_K_ genes [36]. In leaves, *GmIDA2a* and *2b* with ethylene-independent highest absolute and relative AZ expression level at the base of the petioles, are the paralogs most similar to *AtIDA*. Interestingly, GmHAE3b and GmHAE5a/5b with the highest similarities to AtHAE and AtHSL2, respectively, were also adequately expressed in the petiole [36]. Thus, these expression patterns suggest the involvement of the IDA-HAE/HSL2 module in leaf abscission. Interestingly, *GmIDL2a*, as well as the PIP_K_ type *GmIDL4b*, are expressed in the cells overlaying lateral root primordia, and overexpression of the encoded peptides enhances LR emergence, ectopic separation of root cells, and increases the expression levels of EXPs, XTHs, and PGs [37].

The other major group of dicot plants, the Asterids, have, independently of the Rosids, gone through diverse whole genome duplications [28], but also have genes encoding PIP_R_ and PIP_K_ peptides, consistent with the identification of IDA and IDL4 homologues in eudicots that evolved before the Rosid-Asterid split. For instance in tomato (*Solanum lycopersicum*) there are one PIP_K_ and several PIP_R_ genes, including one expressed in AZ [36].

We have chosen to investigate poplar, a species of the order Malpighiales, which is the most distant order of the Rosids clade compared to the Brassicales. Poplar trees grow in the northern hemisphere and shed their leaves in the autumn in response to lowered temperature and shorter day length. Experimentally, cell separation in the preformed AZ at the leaf axil of hybrid aspen (*Populus tremula X Populus tremuloides*), can be induced by wrapping branches in foil to keep the light out, thus mimicking the dark and long autumn nights [42]. A significant upregulation of *PpIDA* and *PpIDL1* was seen in in these AZs [18].

Fall of whole flowers occurs both in monocot and eudicot species, when pollination or fertilization fail [43]. Abscission of immature fruit is a normal event in several cultivated crops [38,39]. For instance, in citrus (*Citrus sinensis*), of the order Sapindales of the Rosids, the expression of the gene encoding the peptide most similar to AtIDA (*CitIDA3*) could be correlated to the process of fruit abscission [39]. Interestingly, *CitIDA3* promoter coupled to the GUS marker gene and transformed into Arabidopsis, was expressed in the floral organ AZs in flower positions 5–8 counting from the top of the inflorescence, the same positions where *pAtIDA:GUS* is expressed [39]. This suggests that genes expressed in the AZ cells of different organs in different species have conserved promoter elements recognized by conserved transcription factors. Interestingly, important downstream components of the IDA-HAE/HSL2 pathway, like the KNAT transcription factors and cell wall remodeling genes, have been identified in both Arabidopsis and Citrus [25].

Litchi (*Litchi chinensis*) is another fruit tree where abscission is a problem. From the first flowering there are waves of flower abscission after bloom. The loss is unpredictable, however, starting from 60,000 flowers per tree less than 50% develop mature fruits. Three genes encoding IDA/IDL homologues have been identified, whereof the *LcIDL1* expression level increases as the abscission process progresses in litchi floral AZs [38]. As for CitIDA3, the PIP motif of LcIDL1 is identical to that of Arabidopsis PIP_R_ except for the Ala-Gly switch in position 6. There is also a conservation of the flanking amino acid residues that are likely recognition sites for endopeptidases which process the preprotein to a mature protein. The peptides encoded by the two other *LcIDL* genes are more similar to AtIDL4 and AtIDL5, and their flanks differ, suggesting that they are processed by different endopeptidases.

A further indication of a conserved role of LcIDL1 in abscission was provided by rescue of the abscission deficient phenotype of the Arabidopsis *ida* mutant using *LcIDL1* expressed under the promoter of *AtIDA* [38]. Additional substantiation of the function in cell separation was demonstrated by an increase in cytosolic pH in AZ cells, and induction of CWR genes previously shown to be regulated by IDA-HAE/HSL2 in Arabidopsis [38].

As for *CitIDA3*, *35S:LcIDL1* transformed into Arabidopsis resulted in a phenotype very similar to that of *35S:AtIDA*, with early abscission [34,38,39]. However, the authors report that when using a *35S:LcIDL1-YFP* construct transformed into protoplasts, the YFP appears in the cytosol. It is not expected that an IDA peptide can work in the cytosol, as the peptide must be transported to the apoplastic space to bind the ectodomain of the receptors. Therefore, we find it unlikely that the location of the YFP reflects the location of the active peptide. Instead, after processing on both sides of the PIP motif, the mature peptide could possibly be secreted while the C-terminal amino acids fused to YFP may remain in the cytosol. Alternatively, appropriate processing enzymes may not be expressed in leaf protoplasts.

Our phylogenetic investigation identified a PIP_R_ version of the IDA homologues also in monocots, for example, in oil palm (*Elaeis guineensis*) of the order Arecales. Dicots and monocots diverged about 130 Myr ago, and thus, the conservation of the peptide suggests a very strong natural selection against changes in the peptide sequence. We have reported an upregulation of *EgIDA* genes in ripened AZ of the oil palm fruits [18]. As in many other species, ethylene is produced at the onset of the abscission process, and contributes to the necessary maturation of the AZ cells [17,41]. In oil palm, ethylene is a prerequisite for abscission to take place and *EgIDA5* is induced by exogenous ethylene. AtIDA was launched as a factor representing an ethylene-independent pathway of abscission in the sense that the *ida* mutant phenotype cannot be rescued by exposure to ethylene, and does not display phenotypes, for example, hypocotyl elongation of dark-grown seedlings, typical for mutations in genes involved in perception of ethylene [10]. However, in Arabidopsis, ethylene is also produced in mature AZ when *AtIDA* is expressed [44].

#### 2.2.3. Matching Ligands and Receptors

Our phylogenetic investigation has also addressed the ancestry of the receptors HAE and HSL2 [18]. With the exception of the grasses, receptors most similar to HSL2 were found present in all orders investigated, indicating that an orthologue of HSL2 existed already when the flowering plants emerged. HAE is, on the other hand, the likely result of a genome duplication that took place in early dicots [18]. The upregulated expression of IDA genes in the AZ in several species, albeit in different organs, is consistent with a conserved function in regulation of cell separation processes. This requires AZ expression of receptors that are conserved both regarding downstream output from the kinase domain upon receptor activation and with respect to peptide and co-receptor binding. SERK1, a co-receptor identified in all land plants [45], in Arabidopsis interacts with IDA and HAE [22], and SERK1-interacting residues, Arg-His-Asn (RHN), at the C-terminal end of PIP_R_ as well as the amino acids involved in co-receptor binding in HAE/HSL2, are also nicely conserved in the receptors discussed here (Appendix A).

The crystal structures of AtIDA and AtIDL1 synthetic peptides bound to AtHAE have recently been solved, and revealed that the ligand is positioned along the inner surface of the LRR ectodomain of the receptor [22]. The residues directly interacting with the amino acids of the peptide are among the best conserved, apart from the leucine (Leu, L), Asn, Gly, isoleucine (Ile, I) and Pro residues that make up the prototype scaffold of the 24 amino acid long LRR units (LxxLxxLxLxxNxLSGxIPxxLGx where x is any amino acid). Using the AtHAE-AtIDA crystal structure (PDBid:5ixq) [22] and homology modelling (SWISS-model) [46], three-dimensional structures were generated for the ectodomain of AtHSL2, and the HSL2 othologue from *E. guineensis* (EgHSL2). [18]. The interactions between these modelled receptors and the respective IDA peptides were investigated. Consistent with binding and activation data [21], there is a high degree of conservation between AtHAE and AtHSL2, likely to facilitate strong interaction between AtIDA and the AtHSL2 receptor (Figure 4a), and most peptide-receptors hydrogen bonds seem to be preserved (Figure 4b). The COO^-^ group of Asn in the AtIDA peptide forms hydrogen bonds with Arg407 and Arg409 in AtHAE [22]. However, Arg407 is replaced with Tyr415 in AtHSL2, likely weakening the interaction with the C-terminal Arg-His-Asn motif in AtIDA. Nonetheless, the hydrogen bonding network in the vicinity of the central hydroxylated Pro (Hyp) of AtIDA is likely strengthened in the AtHSL2-AtIDA model, compared to in the AtHAE-AtIDA crystal structure, due to the presence of an Asn residue (Asn296) in AtHSL2, replacing Arg288 in AtHAE (Figure 4a,b).

The overall interaction between the modelled EgHSL2 receptor with the corresponding EgIDA peptide (Figure 4c) seems to be weakened compared to the interactions seen for the Arabidopsis AtHAE-AtIDA crystal structure and AtHSL2 model. It is, however, still likely to be sufficient for a receptor-peptide binding interaction (Figure 4d) if the central Hyp is replaced with a non-hydroxylated proline in the EgIDA peptide. A weaker hydrogen bonding network in this region is likely in this species, although a probable hydrogen bond is formed between the nearby Trp272 residue in EgHSL2 (Phe268 AtHAE) and the carbonyl group of Gly residue in EgIDA, strengthening the interaction with the peptide.

To investigate the interaction between the EgHSL2 receptor and the corresponding EgSERK1 co-receptor, a homology model of the EgHSL2-EgIDA-EgSERK1 signaling complex was generated (SWISS-model), using the Arabidopsis AtHAE-AtIDA-AtSERK1 complex structure (PDBid:5iyx) as a template [22]. A structural alignment of the model and structure is shown in Figure 5. Residues on the SERK1 co-receptor, previously shown to be involved in hydrogen bonding to the HAE receptor, are conserved in AtSERK1 and EgSERK1 (Appendix A). Also, a high degree of conservation is shown in the HAE/HSL2-SERK1 interface in the *E. guineensis* model complex, as compared to the *A. thaliana* crystal complex. As also observed from the EgHSL2-EgIDA interaction (Figure 5), although fewer hydrogen bonds are suggested between EgHSL2 and EgSERK1 from the predicted EgHSL2-EgSERK1 model, the interaction is still likely sufficient for complex formation.

## 3. Conclusions and Perspectives

Starting with the identification of the different components of the IDA-HSL signaling pathway in Arabidopsis, we and others have now identified orthologues of these genes in all orders of flowering plants; provided examples of their expression in AZs or flowers, leaves and fruits; demonstrated that IDA peptide can induce the abscission processes both in monocots and dicots; and in a few cases validated the hypothesized conserved function across organ and species by complementation of the Arabidopsis *ida* mutant. The ultimate proof of conserved function is however still lacking—a total deficiency in abscission by mutation of the peptide or receptor genes has so far only been shown in Arabidopsis.

Our lab initiated the study of the *ida* mutant about 20 years ago due to its distinct abscission—deficient phenotype. Over these years technological innovations have revolutionized science, including plant science, so, in principle, we can access the genetic information of any species, any individual and even any cell. Whole genome sequencing of species from every order of flowering plants makes it possible to study variation in morphology, development and environmental adaptation from an evolutionary perspective.

Peptides have been discovered as novel hormones facilitating cell-to-cell communication both in relation to developmental processes and defense (see recent reviews [47,48,49,50]). IDA-HAE/HSL2 is among the best elaborated peptide-ligand receptor modules per date, with genetic, biochemical and structural evidence for peptide-receptor interaction, identification of co-receptors, and components of a downstream signaling pathway with MAP kinases and KNOX transcription factors, controlling the expression of genes involved in the actual cell-separation step [4,5,6,10,13,14,21,23,25,26,34,35]. One fundamental question in evolutionary biology is whether evolution of traits is mainly a result of mutational changes in proteins or changes in the regulation of when and where genes are expressed. We hypothesize that with respect to abscission processes the latter is the case; the molecular basis for cell separation is conserved in the IDA-HSL ligand receptor signaling module no matter where and when organ abscission takes place.

Farmers suffer severe losses due to untimely abscission of flowers and fruits, and higher yield per acre would also be welcome from an environmental perspective. With the adaptation, in a steadily increasing number of plants species, of the CRISP/Cas gene editing technique that promises targeted mutagenesis of any gene, we anticipate that our hypothesis will be tested by generation of *IDA* or *HSL* mutants in relevant species in the near future.

## Figures and Tables

**Figure 1 plants-08-00225-f001:**
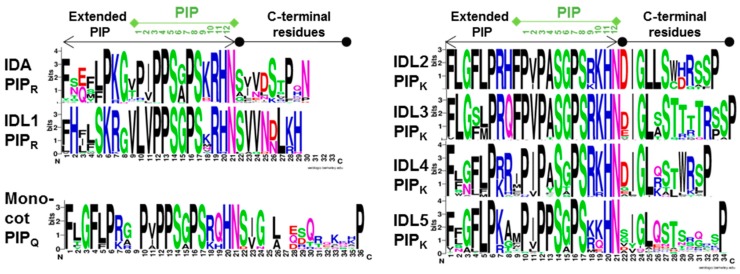
Conserved residues in propeptides of angiosperm IDA and IDL1-5 orthologues numbered from the start of the extended PIP motif to the C-terminal end. Sequences were aligned using Muscle, and consensus sequences were generated with logo@compbio.berkeley.edu. The PIP residues are numbered 1–12. IDL2-5 deviate from the IDA and IDL1 orthologues with a Lys (K) instead of an Arg (R) residue in position 10. IDL2, 3 and 4 have typically an Ala (A) instead of a Pro (P) in position 4. PIP_Q_, which has a glutamine (Gln; Q) in position 10, was frequently found in the grasses (Poales) of the monocots [18].

**Figure 2 plants-08-00225-f002:**
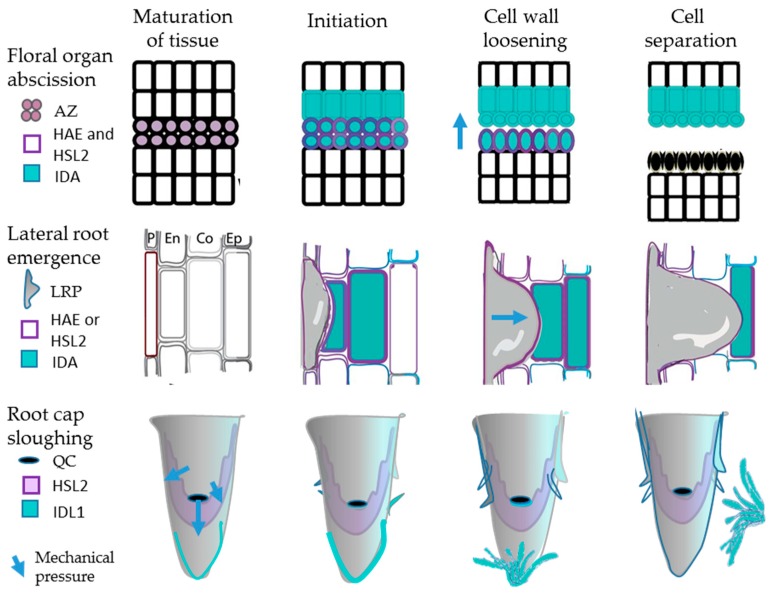
IDA/IDL and HAE/HSL2 involvement in floral organ abscission, lateral root emergence and root cap sloughing. These processes involve maturation of abscission zone (AZ) cells, cells overlying lateral root primordia (LRP), and lateral root cap (LRC) cells, respectively, and gradual cell wall loosening before the actual cell separation takes place. Mechanical pressure from expanded AZ cells in floral organ abscission, the LRP itself during emergence, and new LRC cells and elongating columella cells during root cap sloughing, is likely to contribute to these processes. QC—quiescent center.

**Figure 3 plants-08-00225-f003:**
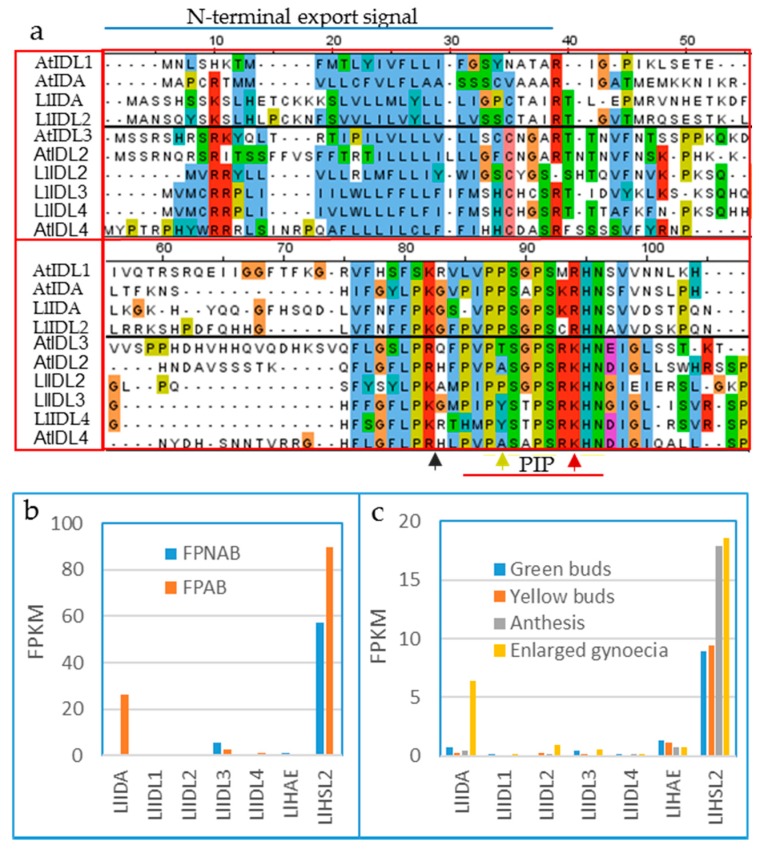
IDA-HAE/HSL2 of yellow lupine (*Lupine luteus*). (**a**) Multiple sequence alignment (generated with Jalview version 2) of prepropeptide sequences from Arabidopsis (AtIDA/AtIDL) and yellow lupine (LlIDA/LlIDL), colored with ClustalX color scheme (blue—hydrophobic; red—positive charge; mangenta—negative charge; green—polar; cyan—aromatic; cysteine—pink; proline—yellow; and glycine—orange). The N-terminal export signal and the PIP motif are indicated. Red arrow—position 10 of the PIP motif, either R or K; yellow arrow—position 4 of PIP, P often mutated in PIP_K_ peptides; black arrow—suggested processing site for endopeptidase. Transcription level of *LlIDA*/*LlIDL* and *LlHAE*/*LlHSL2* in (**b**) non-abscising (FPNAB) and abscising (FPAB) floral pedicels and (**c**) during four stages of flower development, identified by RNA sequencing (FPKM—Fragment Per Kilobase of exon per Million fragments mapped). Data was extracted from BioProject ID PRJNA419564). LlIDL1—OIV90565.1; LlIDL2—OIW07821.1; LlIDL3—OIW17824.1; LlIDL4—OIV98063.1; LlIDL5 AMH85930.1; LlIDA TRINITY_DN45322_c0_g2_i1, CP023114.1:17426727-17426888; LlHSL2—TRINITY_DN55300_c2_g2; LlHAE—TRINITY_DN57817_c0_g1.

**Figure 4 plants-08-00225-f004:**
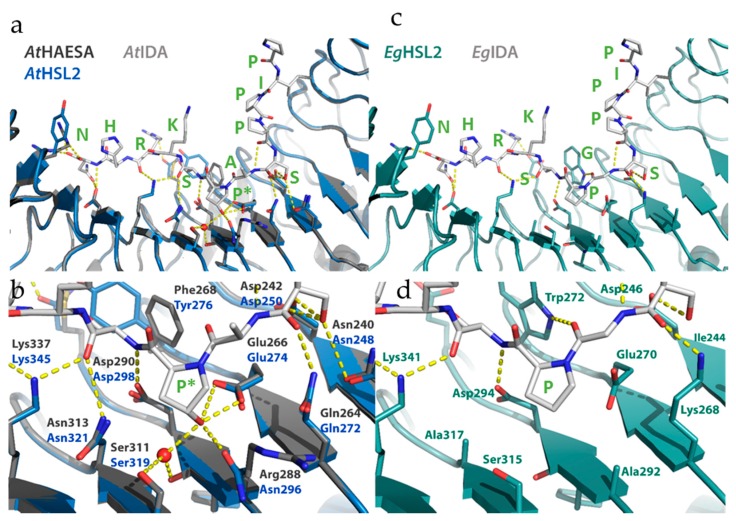
Structures of AtHAE, AtHSL2, and EgHSL2, with bound AtIDA and EgIDA peptides. (**a**) The AtIDA binding pocket covers leucine-rich repeats (LRRs) 2–14 and all residues originate from the inner surface of the AtHAE superhelix. Overall alignment of the AtHAE structure (PDBid:5ixq) (dark grey) with the AtHSL2 model (blue), with AtIDA peptide bound (light grey), with (**b**) close-up view of the central Hypregion. (**c**) Overall model of EgHSL2 (teal) with EgIDA peptide (light grey). with (**d**) close-up view of the latter structure and models showing the central parts in EgHSL2 with the surrounding hydrogen bonding network. Central amino acids of the receptors, as well as the IDA peptides, are shown as sticks and colored by atom type. Water molecules are shown as red spheres. Hydrogen bonds are depicted as dotted lines (yellow). Residues involved in hydrogen bonding to the peptide are depicted with three-letter symbols in colors according to the respective structures, as in (**a**,**c**). The peptide residues are shown in green wone-letter symbols. P*—hydoxylated Pro. All structure figures were prepared using PyMOL (Schrödinger, LLC).

**Figure 5 plants-08-00225-f005:**
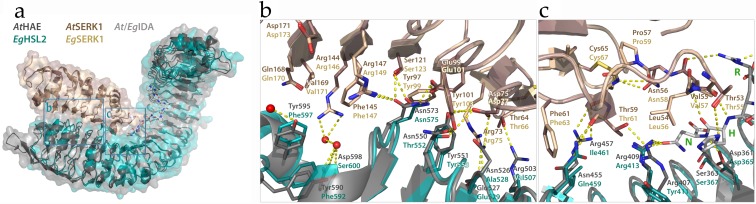
Structural alignment of the of *A. thaliana* HAE-IDA-SERK1 crystal complex (PDBid:5iyx) and the *E. guineensis* HSL2-IDA-SERK1 model complex. (**a**) Overall alignment of the two structures in cartoon and surface representation, with IDA shown as sticks. (**b**) Alignment of the hydrogen bonding network in the HAE/HSL2-SERK1 core interface and (**c**) in the HAE/HSL2-IDA-SERK1 interface in both complexes. Central amino acids, as well as the IDA peptides, are shown as sticks and colored by atom type. Water molecules are shown as red spheres. Hydrogen bonds are depicted as dotted lines (yellow). Residues involved in hydrogen bonding are depicted in colors according to the respective structures, as specified in (**a**). All structure figures were prepared using PyMOL (Schrödinger, LLC).

**Table 1 plants-08-00225-t001:** Overview of IDA/IDLs during abscission and other cell separation processes in plants.

Genes	Process	Methods ^1^	Species
*IDA, HAE, HSL2*	Floral abscission	Mutant; 35S-phenotype; Binding assay; Crystal structure	Arabidopsis [10,21,22,34]
*IDA, HAE, HSL2*	LR emergence	Mutant	Arabidopsis [5]
*IDA, HAE, HSL2*	Leaf abscission	Mutant	Arabidopsis [4,35]
*IDL1, HSL2*	Root cap sloughing	Mutant; Enhanced-phenotype; Binding assay	Arabidopsis [6]
*GmIDA2a/2b,* *GmHAE3b/5a/5b*	Leaf abscission	Expressed in AZ	Soybean [36]
*GmIDL2a/4b*	LR emergence	35S-phenotype	Soybean [37]
*LcIDL1*	Flower/Fruit abscission	Expressed in AZ; 35S-phenotype; Rescue of *ida* mutant	Litchi [38]
*CitIDA3*	Fruit abscission	Expressed in AZ 35S phenotype Rescue of *ida* mutant	Citrus [39]
*LlIDA*	Flower abscission	Expressed in AZ; Synthetic peptide	Lupin [40]
*EgIDA2/5*	Fruit abscission	Expressed in AZ Synthetic peptide	Oil palm [18,41]
*PpIDA, PpIDL1*	Leaf abscission	Expressed in AZ Synthetic peptide	Poplar [18,41]

^1^ Only main approaches are listed for studies in Arabidopsis.

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
