# Peer review of "Control of Organ Abscission and Other Cell Separation Processes by Evolutionary Conserved Peptide Signaling"

_plants, 2019, doi:10.3390/plants8070225_

Round 1
Reviewer 1 Report
All included in Comments & Suggestions PDF file

Author Response
MDPI reference
Control of organ abscission and cell separation
processes by evolutionary conserved peptide
signalling
Chun-Lin Shi, Renate Marie Alling, Marta Hammerstad and Reidunn B. Aalen*
This review on peptide/receptor signalling in controlling flower abscission (and lateral root emergence) and the view on the evolutionary conservation of their function in closely and more distantly-related species is very informative & coming from the leading group in this field of abscission signalling in specialised tissue and/or cell types in higher plants. A good overview on the current status of this research is presented including – as expected from the title – a nice comparative evolutionary analysis on IDA/IDL peptide - HAE /HSL2 membrane receptor complex.
Although I am not involved in this highly specialized area of cell-type specific signalling during abscission, this review has been well written with good referencing, still some knowledge, basics and area related terms are not well described to make this understandable for a broader plant research community. The comments given below may help the authors to give directions for improvement of a few important details. The addition of diagrams on how – and in which sequence – the signaling during both, flower abscission and lateral root emergence, would help understand all the different gene names/annotations. Overall a joy to read.
A new figure has been included as to show how the three processes relate.
Comments (from this reviewer in italics)
Abstract
Cell separation is in function during opening of anthers to release pollen; flowers or floral organs are
detached when they have served their purpose in fertilization; fruits and seeds are abscised from the
mother plant 12 to secure the propagation of new generations.
· Cell separation, see below (introduction) and –incorrect- whole flowers only senesce if not
successfully fertilised (see also authors own comment sentence 264-265), thus not served
their purpose in fertilization.
Changed into, floral organs are detached after pollination when they have served their purpose; unfertilized flowers are shed;
… genes encoding cell wall remodelling proteins ………….
· What are (do? functionally?) remodelling proteins, a have to come back to this in Intro
Changed into, encoding enzymes involved in cell wall remodelling and disassembly
IDA encodes a small peptide
· One IDA, one peptide? There are multiple IDA orthologous genes encoding IDA peptides?
Yes, one IDA and 8 IDLs in Arabidopsis, and there are multiple IDA orthologous genes in other flowering plants
cell separation processes in all kind of species and organs.
· Suggestion: change into ‘angiosperms’ or ‘flowering plant species’
Changed into, involvement in various cell separation processes in angiosperms.
1. Introduction
General question: the definition of abscission and cell separation is/are not very clear in case of AZ (flower) and/or lateral root development? When authors specifically use these terms to differentiate between (I) floral abscission involving the AZ, and (II) cell separation in the case of lateral root protrusion through outer layers, I do understand, but should be clarified. If not, and these processes are ‘similar’, then (suggestion) I would swap both terms in the title, because cell ‘separation’ (see also below) comes before the actual abscission.
Changed into, organ abscission and other cell separation processes, in the title.
If the two processes are regarded as ‘different’ (which is the case to my opinion) – extremely interesting involving one peptide/receptor complex resulting in two different responses - then why is the resultant of both different (separation vs abscission)? Furthermore, do authors know what happens with the initial ‘separating’ cells and wider surrounding cells, while the lateral root increases its diameter during development into mature lateral roots: do these ‘abscise’ eventually?
Cell separation ? for wider Plant Research community.
· What do authors mean by cell "separation"? Physically, structurally, biochemically?
Cell separation (broadly) defines all cell separation (opposite to cell adhesion) processes including organ abscission, sloughing, cell-cell opening to form gap/hole, and so on. While abscission often defines cell separation processes that abscise/shed whole organs such as leaf, flower, fruit and floral organs.
To avoid confusion, we changed into organ abscission and other cell separation processes when we both are mentioned/ compared with each other
.
When pods mature and dry, tension builds up between a lignified dehiscence zone and a
separation layer, and this leads eventually to opening of the valves
· Comment: similar in during anther dehiscence, specialised linage of cells: stomium and
Endothecium
Changed into, “and displays opening of anthers and siliques (so-called dehiscence) to release pollen and drop seeds [2]. Dehiscence relies on specialized tissues that breaks open when tension builds up upon dehydration [3].”
· Explain ………. dissolution? loosening?
Changed to degradation
of the signalling peptide IDA and its receptors
· Single peptide with more than 1 receptor?
Yes, single peptide with 2 receptors
2. The role of IDA/IDL peptides during organ abscission and cell separation processes
bop1 bop2
· Explain “bop”
Explained, BLADE-ON-PETIOLE1/2
fail to export a number of 67 proteins needed for middle lamella dissolution in the AZ (Refs)
· (Refs needed)
Added
Of the many LRR-RLK genes in Arabidopsis ..,
· Meaningless ………….. How many, number?
Changed into, Of the approximately 235 LRR-RLK genes in Arabidopsis
were found to be expressed in the floral organ AZs, and the hae hsl2 double mutant was, like ida,
totally deficient in abscission ….
· Demonstrated IDA binds to HAE /HSL2 ?
Here we describe the similar mutant phenotype in ida and hae hsl2, only indicating they are potential ligand and receptors. The detailed binding is described below in IDA and induction of abscission part (Biochemical and structural evidence has substantiated IDA as ligand for HAE and HSL2 [21,22]. Activation of HSL2 results in an oxidative burst, and an assay where synthetic IDA peptides of different concentrations and lengths were added to tobacco leaf pieces expressing HSL2, demonstrated that the 12 amino acids of PIP with hydroxylated proline (Pro, P) in position 7 could bind and activate the receptor efficiently at biologically relevant nanomolar concentrations, hundred times more efficiently than EPIP. The histidine (His, H) and Asparagine (Asn, N) in positions 11 and 12 seem essential for function, but the amino acids C-terminally to Asn can be removed. Synthetic hydroxylated IDA PIP peptide (Figure 1) bound to the HAE-LRR was used to obtain the molecular structure of this ligand-receptor pair and their interaction with the co-receptor SERK1 [22].)
thus indicating that IDA is regulating some cell wall remodelling genes, consistent with microarray and transcriptome data [16].
· ………… such as …………… specify which ones, and how does this affect process (abscission or separation, or both?)
Changed into, thus indicating that IDA is regulating some cell wall remodelling (CWR) genes. For example is the expression of XYLOGLUCAN ENDOTRANSGLYCOSYLASE6 (XTR6), POLYGALACTURONASE LATERAL ROOT (PGLR) and POLYGALACTURONASE ABSCISSION ZONE ARABIDOPSIS THALIANA (PGAZAT) greatly reduced in ida and hae hsl2 mutants during floral organ abscission , consistent with microarray and transcriptome data [16].
2.2. IDA and induction of abscission
…. the other conditions and factors needed for abscission to take place …..
· Such as ….?
It is described in the following sentence (Formed AZs, expression of HAE and HSL2), Consistent with this, vestigial AZs have been discovered at these sites, and the HAE and HSL2 receptors are expressed there [13,18].
more efficiently than EPIP (Reference!).
· Reference!
Added.
The prepropeptide is thus assumed to be exported to the endoplasmic reticulum, processed, and
secreted to the apoplastic space where it can move and interact with its receptors present in the
plasma membrane.
· Any idea: what signals expression of IDA? (ethylene?), to be transcribed, translated, secreted
- and in turn - interact with the membrane receptor (+ SERK1 see below as well) being
present ….. to trigger cell separation? Why so complicated, why need for secretion first?
The signals that trigger expression of IDA are largely unknown during floral organ abscission, we showed IDA can be induced by auxin during LR emergence.
Although there are no direct evidences, we think such complicated process is necessary to ensure that only the right cells/organs abscised from the main plant.
2.3. IDA and IDL1 are involved in cell separation events in the root
… delay LR emergence in that the LRP has to force 146 itself out, which destroys the cells.
· This has been demonstrated that “mechanical force” is responsible for localised “wounding”
of cells in layers and “destroy” these cells ………?
Corrected
During floral abscission also found LR emergence: …. cell wall remodeling genes, typically expansins
(EXPs), xyloglucan endotransglucosylase/hydrolases (XTHs), and polygalacturonases (PGs) have been
identified through suppressor screens and mutant studies [24,25]
· 24. Niederhuth, C.; Patharkar, O.R.; Walker, J. Transcriptional profiling of the Arabidopsis abscission mutant 492
hae hsl2 by RNA-Seq. BMC Genomics 2013, 14, 37. 493
· 25. Shi, C.-L.; Stenvik, G.-E.; Vie, A.K.; Bones, A.M.; Pautot, V.; Proveniers, M.; Aalen, R.B.; Butenko, M.A. 494
Arabidopsis Class I KNOTTED-Like Homeobox Proteins Act Downstream in the IDA-HAE/HSL2 Floral 495 Abscission
Signaling Pathway. The Plant Cell Online 2011, 23, 2553-2567, doi:10.1105/tpc.111.084608.
Transcription factors working downstream of IDA-HAE/HSL2 during floral organ abscission and cell wall remodeling genes, typically expansins (EXPs), xyloglucan endotransglucosylase/hydrolases (XTHs), and polygalacturonases (PGs) have been identified through suppressor screens and mutant studies [24,25]. Interestingly, a number of the same factors have been found to be involved in separation of cortex and epidermal cells to facilitate LR emergence [26].
The references here in the text are correct, 24 refers the CWR genes in hae hsl2 transcriptome, 25 refers the downstream TF BP/KNAT1 identified from ida suppressor screen. While 26 refers the common factors MPK3/6 and MKK4/5 involved in LR emergence. We also added our LR emergence paper here (5. Kumpf, R.P.; Shi, C.L.; Larrieu, A.; Sto, I.M.; Butenko, M.A.; Peret, B.; Riiser, E.S.; Bennett, M.J.; Aalen, R.B. Floral organ abscission peptide IDA and its HAE/HSL2 receptors control cell separation during lateral root emergence. Proc Natl Acad Sci U S A 2013, 10.1073/pnas.1210835110, doi:10.1073/pnas.1210835110.) to refer the CWR genes during LR emergence.
2.4. IDA/IDL1 and HAE/HSL2 peptide receptors are present in all angiosperm orders
…….. we collected putative IDA/IDL sequences
· At this point still not clear how many IDA/IDL genes in Arabidopsis
We changed a sentence in 2.1. to clarify this:
IDA, on the other hand, is member of a small family of 9 genes encoding IDA and IDA-LIKE (IDL) preproproteins of less than 100 amino acids with N-terminal export signal
2.5. Are IDA/IDL and HAE/HSL involved in cell separation in other species than Arabidopsis?
in particular in crops where abscission is a problem
· Such as ……… examples?
Changed into, in particular in crops such as citrus species and litchi where abscission is a problem
CWR genes are activated
· Proteins get activaten, …….. genes get transcribed ……… or show expression ….
Changed into, CWR genes are transcribed
Compared to enzyme-encoding genes, signalling peptides …….. may be less abundant and may
therefore be overlooked ….
· One cannot compare “genes” with “proteins/peptides”. … less abundant? Transcripts and/or
protein/peptides?
Changed into, Compared to that of enzyme-encoding genes, transcription of signalling peptides genes may be less abundant
However, the underlying data the on IDA, IDL and 217 HSL transcript levels
· Rewrite
The whole paragraph has been rewrittten
Throughout text and now alignment in Figure 2 / Legend: …….. yellow lupin (Lupine luteus). (a)
Multiple sequence alignment of IDA 254 and IDL preprosequences from Arabidopsis and yellow lupin
· Confusing: Why no consistency in gene names including Arabidopsis thaliana: add At .. for
example AtIDA/1-77 and AtIDL1/1-86, as Ll for Lupin, Pp, Gm, Lc. Eg etc
Whereas in Figure 3, authors do make use of At to clarify: Figure 3. Structures of A. thaliana HAE, A.
thaliana HSL2 … AtHSL2 and AtIDA?
Gene names are changed properly when different species are mentioned to avoid confusing.
Additional indications of the function in cell separation were demonstrated by an increase in
cytosolic pH in AZ cells, and induction of CWR genes previously shown to be regulated by IDA-HAE/
HSL2 in Arabidopsis [40].
· Mentioned, but what is functional significance of pH increase?
This is a good question. But as far as we know, it is not clear.
….. to that of 35S:IDA, with early abscission
· Presumably authors mean 35S:AtIDA (from Arabidopsis) – see also previous comment on use
of At
Yes, and changed into, 35S:AtIDA
…. remain inside the protoplasts
· Suggestion: remain in the protoplast cytoplasm?
· Any data on the trafficking and subcellular localisation of prepro/IDA protein
endopeptidases? and in addition, cell wall modifying enzymes?
Has been addressed
IDA in Arabidopsis……………
· AtIDA ….?
Changed
HSL2 was found to be the oldest …………
· What do authors mean by “to be the oldest”? of what?
Changed text
This implies that HSL2 was present at the time when the angiosperms arose
Structural studies on interasctions (page 9-10):
SERK1, a co-receptor identified in all land plants [44], is in Arabidopsis interacting with IDA and HAE
[22],
· Any indication for a putative role for Brassinosteroids? In this signalling pathway
(counteraction? …)
We do not wish speculate in this direction
· Figure 4 and text, including “a homology model” should be emphasised – dealing with
prediction, has not been demonstrated. Sentences as ….. “the EgHSL2-SERK1 interaction
seems to be somewhat weaker” sounds very hypothetical / weak …
Changed into, although fewer hydrogen bonds are suggested between EgHSL2 and EgSERK1 from the predicted EgHSL2-EgSERK1 model, the interaction is still likely for sufficient complex formation
….. demonstrated that IDA peptide can speed up the abscission processes
· What do authors mean by “speed up” process?
Changed into, induce
Peptides have been discovered as novel hormones facilitating cell-to-cell communication both in
relation to developmental processes and defence.
· Missing reference(s)
Added
A few consideration:
What happens at the interface between AZ zone and neighbouring tissue(s)? When this “Separation Signalling” is triggered?
What about protein turnover and/or delivery and potentially removal from HAE /HSL2 receptors from target – localized - membranes (such as found for PIN proteins/auxin signalling)?
Plants (journal): as mentioned earlier, missing a clear picture/diagram of IDA signalling (pathway)
and all current key players involved in different tissues (AZ and lateral root emergence).
A new figure has been provided to give a comparison between floral organ abscission, lateral root emergence and root cap sloughing.
Reviewer 2 Report
The authors reviewed the IDA-HAE/HSL2 signalling pathway during organ abscission and cell separation processes and illustrated their hypothesis of its involvement in cell separation in all kinds of species and organs from an evolutionary aspect. Generally speaking, this article is well-written. The authors performed some additional analyses to support their hypothesis. However, I have a few concerns that are listed below.
1.organization:
a. I would recommend the authors to sub-categorize section 2 into two parts. The first part (2.1-2.3) introduces IDA-HAE/HSL2 signalling pathway, while the second (2.4- 2.6) discusses from an evolutionary aspect.
b. It would be a good idea if the authors could provide a figure as an outline/summary for this review. It is hard to get the flow of this review while reading it.
2.methods:
a.Their figure supplemental 1b is lack of a control. In addition, I am not convinced of their statement on ‘no negative correlation between turgor loss and abscission’ (L92), as the authors did not clearly explain where the pBS value is derived and how is it comparable to their experiments.
b.The authors need to provide more details on how they performed multiple sequence alignment from RNA-seq reads. Figure 2a, which panel refers to Arabidopsis/ yellow lupin? Figure 2c, what does each color represent? The authors would also need statistical tests to support differential gene expression.
3. Validation of the findings
a. 2.4 was named ‘IDA/IDL1 and HAE/HSL2…’. However, the authors did not talk about the latter at all.
b. the authors stated that the Arabidopsis IDA is sufficient for induction of abscission, thus other conditions and factors are needed for abscission to take place (L99). The cause and effect seems to conflict with each other.
c. The discussion on IDL6-7 is abrupt (L102). What point does the authors intend to make?
d. I did not see the ‘correlation between the aa motifs’ presented on L188. The authors need to elaborate on that.
4.Minor points:
L46, should be ‘drought’.
L197, it would be better if the sub-title is declarative instead of a question.
L204, what is ‘CWR’?
Table 1, please add separator to different rows.
Author Response
Comments and Suggestions for Authors
The authors reviewed the IDA-HAE/HSL2 signalling pathway during organ abscission and cell separation processes and illustrated their hypothesis of its involvement in cell separation in all kinds of species and organs from an evolutionary aspect. Generally speaking, this article is well-written. The authors performed some additional analyses to support their hypothesis. However, I have a few concerns that are listed below.
1.organization:
a. I would recommend the authors to sub-categorize section 2 into two parts. The first part (2.1-2.3) introduces IDA-HAE/HSL2 signalling pathway, while the second (2.4- 2.6) discusses from an evolutionary aspect.
b. It would be a good idea if the authors could provide a figure as an outline/summary for this review. It is hard to get the flow of this review while reading it.
We agree with this viewpoint, we have renumbered the parts accordingly.
We are also providing a new figure to show the three cell separation processes in Arabidopsis. This will serve as a kind of summary for the first part. .
Tabel 1 serves well as a summary of the second part.
2.methods:
a.Their figure supplemental 1b is lack of a control. In addition, I am not convinced of their statement on ‘no negative correlation between turgor loss and abscission’ (L92), as the authors did not clearly explain where the pBS value is derived and how is it comparable to their experiments.
We have added the wild type control to the Supplementary Figure1, and adjusted the text.
Thus, there is no strict correlation between turgor loss and failure to abscise, and also no strict correlation between cell expansion and abscission.
b.The authors need to provide more details on how they performed multiple sequence alignment from RNA-seq reads. Figure 2a, which panel refers to Arabidopsis/ yellow lupin? Figure 2c, what does each
color represent? The authors would also need statistical tests to support differential gene expression.
We are sorry that the figure was incomplete, and now provide a revised version. Furthermore the text has been extended and revised:
3. Validation of the findings
a. 2.4 was named ‘IDA/IDL1 and HAE/HSL2…’. However, the authors did not talk about the latter at all.
We have added the relevant sentences regarding HSL2
b. the authors stated that the Arabidopsis IDA is suent for induction of abscission, thus other conditions and factors are needed for abscission to take place (L99). The cause and effect seems to conflict with each other.
We wrote: Thus the other conditions and factors needed for abscission to take place, must be present. Consistent with this, vestigial AZs have been discovered at these sites, and the HAE and HSL2 receptors are expressed there
IDA is the only factor that is lack at these positions. The cause (addition/miss-expression of IDA at these positions) will lead to the effect (organ abscission at these positions).
c. The discussion on IDL6-7 is abrupt (L102). What point does the authors intend to make?
We intend to make the point that IDL peptides are not necessarily involved in cell separation
d. I did not see the ‘correlation between the aa motifs’ presented on L188. The authors need to elaborate on that.
We have changed the sentence and exemplified what we mean.
4.Minor points:
L46, should be ‘drought’.
L197, it would be better if the sub-title is declarative instead of a question.
L204, what is ‘CWR’?
Table 1, please add separator to different rows.
drought is corrected
sub-titles are changed
CWR is short for cell wall remodelling, and it is added when it first appears
separator is added in Table 1